# Multi-Task Joint Learning Model for Chinese Word Segmentation and Syndrome Differentiation in Traditional Chinese Medicine

**DOI:** 10.3390/ijerph19095601

**Published:** 2022-05-05

**Authors:** Chenyuan Hu, Shuoyan Zhang, Tianyu Gu, Zhuangzhi Yan, Jiehui Jiang

**Affiliations:** 1School of Communication and Information Engineering, Shanghai University, Shanghai 200444, China; zhangshuoyan@shu.edu.cn (S.Z.); bme20_gty@shu.edu.cn (T.G.); 2Institute of Biomedical Engineering, School of Life Science, Shanghai University, Shanghai 200444, China; zzyan@shu.edu.cn (Z.Y.); jiangjiehui@shu.edu.cn (J.J.)

**Keywords:** syndrome differentiation, multi-task learning, joint learning, deep learning

## Abstract

Evidence-based treatment is the basis of traditional Chinese medicine (TCM), and the accurate differentiation of syndromes is important for treatment in this context. The automatic differentiation of syndromes of unstructured medical records requires two important steps: Chinese word segmentation and text classification. Due to the ambiguity of the Chinese language and the peculiarities of syndrome differentiation, these tasks pose a daunting challenge. We use text classification to model syndrome differentiation for TCM, and use multi-task learning (MTL) and deep learning to accomplish the two challenging tasks of Chinese word segmentation and syndrome differentiation. Two classic deep neural networks—bidirectional long short-term memory (Bi-LSTM) and text-based convolutional neural networks (TextCNN)—are fused into MTL to simultaneously carry out these two tasks. We used our proposed method to conduct a large number of comparative experiments. The experimental comparisons showed that it was superior to other methods on both tasks. Our model yielded values of accuracy, specificity, and sensitivity of 0.93, 0.94, and 0.90, and 0.80, 0.82, and 0.78 on the Chinese word segmentation task and the syndrome differentiation task, respectively. Moreover, statistical analyses showed that the accuracies of the non-joint and joint models were both within the 95% confidence interval, with $p_{value}$ < 0.05. The experimental comparison showed that our method is superior to prevalent methods on both tasks. The work here can help modernize TCM through intelligent differentiation.

## 1. Introduction

Since traditional Chinese medicine (TCM) was incorporated into the latest global medical outline issued by the World Health Organization (WHO), a growing number of scholars have begun engaging in research related to TCM [1,2]. Preventive treatment of disease is one of the core concepts of TCM health theory. It refers to the use of TCM ideas and methods to prevent the occurrence and development of diseases, which plays an important role in the development of public health services [3]. Evidence-based treatment is the basis of TCM, and accurate syndrome differentiation is important for treatment. The eight principles of syndrome differentiation provide a method to analyze the commonality of diseases and form the basis for other methods of syndrome differentiation [4]. In the eight principle of syndrome differentiation, Yin and Yang can be used to summarize the location of the disease, its nature, and condition to provide a firm foundation for subsequent medical judgments (such as methods of treatment) [5]. 

Syndrome differentiation of Yin and Yang deficiency is based on the physiological and pathological characteristics of the Yin and the Yang, and involves analyzing and summarizing a variety of disease-related information that is collected according to four diagnostics for identification [6]. A large amount of critical information on healthcare is buried in unstructured narratives, such as medical records, which makes its computational analysis difficult [7]. Moreover, mastering syndrome differentiation in TCM is a complicated and time-consuming process. Owing to the different qualifications of clinicians, it is also difficult to maintain a stable curative effect in the treatment of specific diseases. This causes environmental and empirical factors to have a significant impact on the results of syndrome differentiation, and in turn leads to inaccurate and unstable diagnosis and treatment [8]. Therefore, it is important to establish an objective and quantitative computer-aided method of syndrome differentiation, and thus to provide pervasive, personalized, and patient-centralized services in healthcare and medicine, so that the ideas of preventive treatment in TCM can be applied to all aspects of community public health services [9,10]. Furthermore, due to the uneven distribution of TCM resources, with the realization of TCM intelligence, the accuracy of TCM syndrome differentiation and the richness of treatment methods for grass-roots doctors can be significantly enhanced, and it can drive the improvement of TCM services at the grass-roots level [11]. The realization of automatic syndrome differentiation of unstructured text includes two important technologies: Chinese word segmentation and text classification.

Words in Chinese are the smallest linguistic unit that can be independently used. Chinese word segmentation aims to segment a complete Chinese sentence into meaningful words. Unlike in English, words in Chinese do not have clear separators between them. Word segmentation is thus an important initial step and a basic module of human–computer natural language-based interaction in Chinese [12,13]. Moreover, the task of TCM text segmentation is challenging because the medical field has many professional vocabularies and there are ambiguities with modern Chinese. 

Therefore, many researchers have studied the task of word segmentation in medical texts. Li et al. [14] used dictionary- and statistics-based methods of word segmentation to segment textual medical records in Chinese, and explored methods of word segmentation suitable for medical texts. Li et al. [15] applied the capsule network (Capsule) to the task of word segmentation in classical Chinese medical books for the first time. To adapt the Capsule structure to the sequence tagging task, they proposed a sliding capsule window that yielded an accuracy of 95% on a public dataset. Xing et al. [16] proposed a framework for Chinese word segmentation in the medical field based on Bi-LSTM with conditional random fields (Bi-LSTM-CRF). They used the multi-task learning framework of transfer learning and high-resource data to improve performance on word segmentation. Yuan et al. [17] proposed an unsupervised method of Chinese word segmentation based on a pre-trained bidirectional encoder representation from transformers (BERT) model, which achieved good performance.

Text classification refers to the automatic classification of text into several designated categories. Intelligent syndrome differentiation of yin and yang deficiency in traditional Chinese medicine can also be abstracted as a problem of classification of the text pertaining to a given condition [18]. 

With the rapid development of machine learning and deep learning algorithms, a growing number of techniques of text classification have been used in modern research on syndrome differentiation in TCM. Li [19] used the subject-related weighting model to classify medical records in TCM. The support vector machine (SVM) was used to study syndrome differentiation in patients of depression, and yielded a high classification accuracy [20]. Zhao et al. [21] proposed exploring the relation between syndromes for viral hepatitis by using manifold ranking (MR). These studies mainly used machine learning algorithms. In the context of the use of deep learning algorithms for syndrome differentiation in TCM, a deep belief network was used to construct a diagnostic model of chronic gastritis syndromes in TCM [22]. Zhu et al. [23] proposed a deep learning algorithm for identifying the damp-heat syndrome in TCM, and Hu et al. [5] used two neural network models to differentiate the Yin and Yang deficiency in TCM. Liu et al. [24] used the recurrent convolutional neural network (RCNN) and the text-based hierarchical attention network (Text-HAN) to establish end-to-end diagnostic models for TCM to identify syndromes of lung cancer. 

It can be seen from the above introduction, some studies on the word segmentation of TCM texts require constructing a TCM corpus, which is labor intensive. Additionally, one of the key steps for text classification in proprietary areas is Chinese word segmentation, but some studies on syndrome differentiation have ignored the important roles of Chinese word segmentation.

Therefore, in this study, we use multi-task learning (MTL) and deep learning to solve the two challenging tasks of Chinese word segmentation and syndrome differentiation. The goal of MTL in machine learning is to exploit the useful information contained in multiple learning tasks to learn a more accurate model for each [25]. A model can be used to share information between tasks and improve the results. From the perspective of machine learning, multi-task learning can be regarded as a form of inductive transfer, which improves model performance by introducing inductive bias. Particularly, multi-task learning includes a variety of internal mechanisms such as implicit data enhancement mechanism, attention mechanism, eavesdropping mechanism, and regularization mechanism to ensure the effectiveness of multi-task learning [26,27]. Meanwhile, many studies have demonstrated the effectiveness of multi-task learning in multiple domains. In the field of image processing, one past study [28] used the idea of MTL to integrate two important tasks of tongue characterization (tongue segmentation task and tongue coating classification task) into one model and proved the effectiveness of this method through experiments. In the field of natural language processing, Gamal et al. [29] proposed a neural network multi-task learning method for biomedical named entity recognition, and conducted comparative experiments between single-task models and multi-task models on 15 datasets. The results show that the multi-task model produces better NER results than the single-task model, and multi-task learning is found to be beneficial for small datasets. 

Consequently, we seek to improve performance in terms of Chinese word segmentation and syndrome differentiation to analyze medical records in an end-to-end manner. For the two crucial tasks considered here, adequate results of Chinese word segmentation can help retain correct, complete, and acquire important semantic information to obtain better results of syndrome differentiation. The results of syndrome differentiation can provide additional features to help identify specific semantic information to improve word segmentation. These two tasks are related rather than independent, which makes them consistent with the idea of MTL. MTL has delivered outstanding performance in many areas, and this motivates us to incorporate it into our research. Our approach fuses bidirectional long short-term memory (Bi-LSTM) with a text-based convolutional neural network (TextCNN) into MTL. We make the following three contributions to the area in this study:(1)Chinese word segmentation and syndrome differentiation are highly correlated, which makes them suitable for MTL. To the best of our knowledge, this is the first attempt to combine these tasks using MTL. A large number of comparative experiments are used to show that our proposed method is superior to prevalent methods.(2)The proposed model fuses two typical deep neural networks, Bi-LSTM and TextCNN, into the MTL for Chinese word segmentation and syndrome differentiation in TCM. This makes the end-to-end analysis of medical records possible.(3)Each label is annotated and checked by three physicians competent in TCM to ensure the reliability and accuracy of the data.

## 2. Materials and Methods

We propose a model that fuses two typical deep neural networks, Bi-LSTM and TextCNN, into the MTL for Chinese word segmentation and syndrome differentiation in TCM. For the overall research of this paper, we first conducted comparative experiments on different loss function optimization strategies, and determined the use of gradient normalization to optimize the weights of joint loss functions. Then, in order to prove the superiority of our proposed model, we conducted comparative experiments of various baseline models for Chinese word segmentation and syndrome differentiation tasks, and compares them with state-of-the-art models. Finally, we also performed joint and non-joint statistical analysis and ablation studies.

In this section, we first describe the proposed model and its three modules, and then introduce the joint loss function. The baseline model of each task and the evaluation metrics and training details of the model are also introduced.

### 2.1. Proposed Method

The overall framework of our proposed model is shown in Figure 1, the segmentation module is used for Chinese word segmentation and syndrome differentiation is carried out by the classification module. An embedding module common to them is used to provide shared information on these tasks. For the input sentences, the vector representation of the sentence is obtained by the embedding module, and then input to the segmentation module and the classification module, respectively, to obtain the corresponding Chinese word segmentation result and the syndrome differentiation results, respectively. The weighted loss of the two tasks is then defined for multi-tasking loss, and the joint optimization is completed by reverse propagation to help two tasks obtain better performance.

#### 2.1.1. Embedding Module

Defined in Python, the embedding module is a simple lookup table for storing fixed dictionaries and size embeddings. This module relies on indices to retrieve word embeddings. The input to the module is an index list and the output is the corresponding word embedding. Through this module, we can obtain the vector representation of the sentence. 

#### 2.1.2. Segmentation Module

To transform the Chinese word segmentation task into a sequence tagging problem, a label is assigned to each character [16]. There are three types of labels—B, I, and O—corresponding to the beginning, middle, and end of words, respectively, and single-word characters. Given a sequence of n characters $X=\{x_{1},\ldots ,x_{n}\}$, the purpose of the Chinese word segmentation is to find the mapping from *X* to $Y^{*}=\{y_{1}^{*},\ldots ,y_{n}^{*}\}$:
(1)$$Y^{*}=\arg\underset{Y\in {ℒ}^{n}}{\max}p(Y|X)$$
where  ℒ={B,I,O}. 

The long short-term memory (LSTM) unit can learn long-term dependencies without retaining redundant contextual information, can perform well on sequence tagging tasks, and is widely used for natural language processing tasks [30]. The Bi-LSTM is composed of LSTM units, has two parallel levels, and propagates in two directions such that it can input features from the past and the future. Therefore, we use Bi-LSTM models to perform Chinese word segmentation. 

The structure of the module is shown in Figure 2. The embedding layer in vector from is fed into the Bi-LSTM network to obtain the spliced feature vectors of the past and the future. We then obtain the probability of each tag (B, I, and O) through the fully connected layer and identify the tag with the maximum probability. In this way, we can obtain the results of tagging the sequence. Reverse matching according to the corresponding positions of the three kinds of tags is then performed to obtain the corresponding results of text segmentation. Chinese word segmentation can then be visualized, which is convenient for its subsequent application.

#### 2.1.3. Classification Module

Convolutional neural networks (CNN) have been used to classify tongue colors [31], identify cracked tongues, and classify pulse signals [32] in TCM. We use the CNN model to classify texts of medical records in TCM, and thus call it TextCNN. Its structure is shown in Figure 3. For a given character vector, multiple convolutions with varying kernel sizes (3 and 5, the green box is 3, and the red box is 5) are used to convolve the embedded vectors in the convolutional layer. Then, the vectors are passed the max-pooling layer to capture the most salient features. Finally, the results of classification are obtained through the fully connected layer to determine whether the given patient has the Yin deficiency syndrome or the Yang deficiency syndrome.

### 2.2. Joint Loss Function

In essence, the loss functions of the Chinese word segmentation task and the syndrome differentiation task are both cross-entropy, although the former differs from the latter as it uses masked cross-entropy to avoid the influence of padding characters. As reviewed in Ref. [33], the loss function of the multi-task model is defined as the weighted sum of the loss functions of different tasks to optimize all the parameters involved in the tasks. These weights are called hyperparameters. Based on the above, our loss function is defined as in Equation (2). L0 and $r_{0}\;$ are the loss function and the corresponding weight of the Chinese word segmentation task, respectively, and L1 and $r_{1}\;$ are the loss function and the corresponding weight of the TCM syndrome differentiation task, respectively. Our training strategy is to minimize Loss*:*(2)$$Loss=r_{0}\times L0+r_{1}\times L1$$

The weights in multi-task loss are determined by grid search method and dynamic tuning method. Dynamic tuning methods include gradient normalization, dynamic weighted average, and uncertainty-based weighted. Weights of the loss function were fixed during training for the grid search method. The choice of fixed weights might have limited the learning of tasks owing to the varying difficulty of learning and progress of different tasks, the weight of the loss function should be dynamically adjusted. Therefore, Zhao et al. [34] proposed an optimization strategy for gradient normalization, where weights are updated according to the gradient loss. Liu et al. [35] proposed a dynamic weighted average optimization strategy where the weights decrease for tasks in which the loss decreases rapidly. Kendall et al. [36] proposed an optimization strategy for uncertainty weighting where the weight of the task with greater uncertainty is smaller, that is, the task with larger noise that is difficult to learn has a smaller weight. 

### 2.3. Baselines

#### 2.3.1. Chinese Word Segmentation

We compared our method with several LSTM-based models, including LSTM, Bi-LSTM, and Bi-GRU. LSTM is a variant of the recurrent neural network (RNN) model [37], which uses three gates, a forget gate, an input gate, and an output gate, for information transmission [38]. The gated recurrent unit (GRU) replaces the forget gate and the input gate in an LSTM with an update gate [39]. The bidirectional LSTM/GRU network is similar in structure to the LSTM/GRU network, with the difference that it has two parallel levels and propagates in two directions. We also conducted comparative experiments with state-of-the-art models, including the BERT, RoBERTa, and XLNet. The BERT is a bidirectional transformer for pre-training models on large amounts of unlabeled textual data to learn a language representation that can be used to finetune specific machine learning tasks [40]. Both XLNet and RoBERTa are performance-enhancing versions of BERT, which have greatly improved both the amount of training data and computer resources [41,42].

#### 2.3.2. Syndrome Differentiation

In contrast to our model, prevalent methods can be divided into two types: (1) One-stage methods. These methods carry out only syndrome differentiation, and ignore the impact of Chinese word segmentation. (2) Two-stage methods. In these methods, the first stage consists of Chinese word segmentation and the second stage involves methods of text classification to classify syndromes. To ensure the comparability and fairness of the experiments, we used the TextCNN model for syndrome differentiation when using the prevalent methods. Moreover, we used Bi-LSTM in the first step for Chinese word segmentation, and applied the result as the input to the second step for syndrome classification. SVM was used as the traditional method, and the commonly used TextCNN and TextRNN were applied as neural network methods. 

Furthermore, similarly to Chinese word segmentation, we also conducted comparative experiments with state-of-the-art models, including the BERT, RoBERTa, and XLNet.

#### 2.3.3. Ablation Studies

We compared a number of representative models with each module of the proposed model through ablation studies to prove the superiority of Chinese medical text segmentation and syndrome differentiation based on joint multi-task learning. 

Among them, the segmentation module selects Bi-LSTM and Bi-LSTM-CRF, and the classification module choose TextCNN and FastText. Bi-LSTM and TextCNN have been introduced in the previous content, so they are not repeated. The Bi-LSTM layer can extract features in the Bi-LSTM-CRF model while the CRF model can consider the pre- and post-dependencies of the tags [43], which helps obtain better results of word segmentation. That is, it indirectly helps obtain better performance on syndrome differentiation. The FastText model contains three layers: an input layer, a hidden layer, and an output layer [44]. The input layer contains a word vector, and the hidden layer is the superposition and average of multiple word vectors, and the result of classification is obtained through the softmax layer.

### 2.4. Dataset

The data used in this study are 1438 medical records from 146 TCM physicians, and their content is authentic and reliable [45]. The inclusion criteria of yin deficiency syndrome and yang deficiency syndrome were based on the theory of traditional Chinese medicine [6]. The records lacking the complete four diagnosis information of patients were excluded, and 1230 medical records were obtained. To ensure validity and accuracy, we preprocessed the text of medical records using text extraction, and solicited three TCM physicians to annotate and inspect the data [6,46]. As shown in Figure 4, it is an example of data annotation, its mainly includes two steps. The first step is to add syndrome type and word segmentation to the original text, which is completed by three physicians. The syndrome type and the sentence are separated by #, and each word segmentation is separated by /. The second step is to convert the sentence into the corresponding character + sequence label + syndrome type, which is completed by the computer, where the sequence label is converted according to the corresponding positions of the three BIO characters. Following this, a total of 1209 medical records were obtained, including 643 cases of Yang deficiency syndrome and 566 cases of Yin deficiency syndrome.

### 2.5. Evaluation Metrics and Training Details

We compared our model with certain basic models with the same parameter settings. The evaluation metrics consisted of accuracy, specificity, and sensitivity. Owing to the close relationship between syndrome differentiation and treatment in TCM, we also used the receiver operating characteristic (ROC) and the area under the curve (AUC) to assess performance.

We randomly spilt the dataset into those for training (60%), validation (10%), and testing (30%) using seven-fold cross-validation. The adaptive moment estimation (Adam) optimizer was used, the number of epochs was 50, and the learning rate was 0.001. During the training process, the model was saved for testing when the loss in the validation dataset was minimal. We then calculated the average value of each evaluation metric as the final result. 

## 3. Results

In order to prove the effectiveness of the proposed model, we conducted a lot of comparative experiments. For the convenience of expression, we use JCS to represent our proposed joint model, that is, the first letter of the Joint Chinese word segmentation and Syndrome differentiation.

### 3.1. Experiments of Different Loss Optimization Strategies

We used performance on the syndrome differentiation task as an example to assess different strategies of loss optimization. The results of a comparison of different optimization strategies are shown in Table 1, where the range of parameters of the grid search method was [0.1, 0.2, 0.3, 0.4, 0.5], and the best experimental results were obtained when the value of $r_{0}$ and $r_{1}$ was set 0.4. Gradient normalization obtained the best performance in terms of syndrome differentiation. We thus used it as the strategy for loss optimization in the subsequent comparative and ablation experiments.

### 3.2. Experiments on Chinese Word Segmentation

The results of different models are shown in Table 2. Compared with the best baseline model of the LSTM series, our model was superior by 2.65%, 3.45%, and 8.11% in terms of accuracy, specificity, and sensitivity, respectively. The BERT model achieved the best performance among the state of the art. However, it took nearly 10 h to train and had 101.68 M parameters. By contrast, the runtime of our model is 10 min, had 0.47 M parameters, and was only about 1% inferior to the BERT on each of the three evaluation indicators. Therefore, it delivered better performance overall on Chinese word segmentation tasks.

### 3.3. Syndrome Differentiation Experiments

We also compared our model with prevalent models in terms of syndrome differentiation. The results of the methods on this task are shown in Table 3. The first and fourth lines of the table show that the two-stage methods outperformed one-stage methods, which proves the necessity of Chinese word segmentation for this task. Moreover, our model was superior to the best two-stage methods on most indicators, with improvements of 3.98% and 8.16% in terms of accuracy and sensitivity, respectively. Furthermore, our method only needs one feature extraction operation, the runtime of our model is only 10 min, which is faster than those two-stages methods, since these two-stages methods need to conduct feature extraction twice.

We drew the ROC curves of the best baseline model and our model, as shown in Figure 5. The curve of our model was higher, and had an AUC value more than 5% larger than that of the baseline model.

The above comparison shows that our model was superior in terms of performance and time consumed. The results of comparisons with state-of-the-art models are shown in Table 4. Although the BERT model obtained the best performance among the three pre-training models, the accuracy of the proposed joint learning model was higher than that of the BERT model. Additionally, The BERT model also required a much longer training time and had 102.27 M parameters. Therefore, our model delivered better performance overall on syndrome differentiation tasks.

### 3.4. Statistical Analysis

To eliminate the influence of accidental factors in the sampling of the test set on the accuracy of the model, we statistically analyzed the accuracy of the non-joint and joint models based on Bi-LSTM and TextCNN. The specific methods used were bootstrap analysis and permutation testing [47,48]. We conducted 1000 bootstrap resampling on the test set to obtain the average accuracy and average difference of each model as well as these values within a 95% confidence interval. The permutation testing eliminated the influence of accidental factors in the sampling of the test set on the accuracy of the two models by measuring whether there were statistical differences between their results. We proposed a null hypothesis (the prediction-related performance of the two models was the same), constructed a test statistic t (the difference in accuracy between the models), and sampled 10,000 times to obtain a histogram of the difference in terms of model accuracy. According to the histogram to obtain the 95% confidence interval, observe whether the statistics t fall within the 95% confidence interval, and calculate the $p_{value}$.

The results of statistical analyses of the non-joint model and our joint model are shown in Table 5. The accuracies of the two models were within the 95% confidence interval, indicating that they were statistically significant. To exclude the influence of accidental factors, the accuracy of the two models was subjected to a permutation test. The table shows $p_{value}$ was less than 0.05, that is, the difference between the accuracy of the non-joint and the joint methods was statistically significant. The joint model, that is, our proposed model, delivered better performance.

### 3.5. Ablation Experiments

The results of comparative experiments on different modules are shown in Table 6. The performance of the joint learning model on syndrome differentiation was usually better than that of the non-joint model. For example, the joint learning model based on Bi-LSTM-CRF and FastText improved the accuracy, specificity, and sensitivity of the results by 1.5%, 0.59%, and 3.15%, respectively.

## 4. Discussion

### 4.1. Principal Results

Through the extensive experiments, our proposed joint learning model achieves superior performance in TCM Chinese word segmentation and syndrome differentiation tasks. 

For Chinese word segmentation tasks, many studies have used the Bi-LSTM model for Chinese word segmentation and demonstrated the superiority of this model [49]. For example, in the Ref. [50], the optimal F1 value of the word segmentation results reached 95.54%. In our experiments, our proposed JCS model outperforms the Bi-LSTM model on all evaluation metrics, yielded values of accuracy, specificity, and sensitivity of 0.93, 0.94, and 0.90. 

For the syndrome differentiation task, many studies have used the TextCNN model for syndrome differentiation and obtained a high classification accuracy. For example, Hu et al. [5] used TextCNN and FastText model to conduct end-to-end syndrome differentiation experiments, and TextCNN model obtained the highest accuracy rate of 92.55%. Apply this model to our research, the classification accuracy of our proposed JCS model far exceeds that of the TextCNN model by about 7%. 

The above discussions all demonstrate that combining Bi-LSTM and TextCNN models through multi-task learning can effectively improves the performance of these tasks. Because the Bi-LSTM model can capture the contextual information of sentences through propagation in two directions, and the TextCNN model can automatically combine N-gram features to capture the semantic information of sentences at different levels, the proposed joint model can capture richer feature representations of sentences. 

Then, combined with the internal mechanism of multi-task learning to analyze the results. The proposed joint model can introduce inductive bias to play the same role as regularization, reducing the risk of model overfitting. For the eavesdropping mechanism, the label of the word segmentation of Chinese medicine text is the sentence sequence tag, which makes it easier for this task to learn the characteristics of words. However, it is not easy to learn this feature in the syndrome differentiation task, and the word feature is very important for this task. The joint learning of the two tasks allows the syndrome differentiation task to eavesdrop on the features of words in the Chinese word segmentation task for better syndrome differentiation performance. Conversely, the label of the syndrome differentiation task is the specific syndrome type of the sentence, and the symptoms of red tongue are more likely to appear in the syndrome of yin deficiency. Therefore, the model can provide additional evidence that “red tongue” is a group of words for the Chinese word segmentation task through the attention mechanism. That is to say, multi-task learning can effectively improve the performance of both tasks.

### 4.2. Limitations and Future Work

Limitations to the work include that the shared embedding layer compresses the knowledge of the two tasks to the same parameter space, which will exist some information loss. Another limitation is that when using gradient normalization strategy to optimize multi-task loss, each iteration requires additional computation of gradients, which affects the speed of training. 

We intend to consider following aspects in future work to improve the proposed method. Studies have shown that multi-task learning is more suitable for small datasets [29]. Therefore, we will collect more medical records to conduct comparative experiments with different data set sizes to verify the above conclusions. Additionally, when there is enough medical record, we can integrate the BERT model into multi-task learning to improve the performance of the proposed joint model.

## 5. Conclusions

Chinese word segmentation and syndrome differentiation are highly correlated, which makes them suitable for multi-task learning. We proposed a method here to perform word segmentation on textual medical records and classify them. We merged two classic deep neural networks (Bi-LSTM and TextCNN) into MTL to simultaneously conduct these two tasks. To the best of our knowledge, this is the first attempt to combine these tasks using MTL. We compared the proposed method with prevalent methods in the area through a number of experiments. The results showed that it is superior to other methods on both tasks. That is to say, we developed an objective, quantitative, and computer-assisted method of syndrome differentiation that can help modernize traditional Chinese medicine. 

## Figures and Tables

**Figure 1 ijerph-19-05601-f001:**
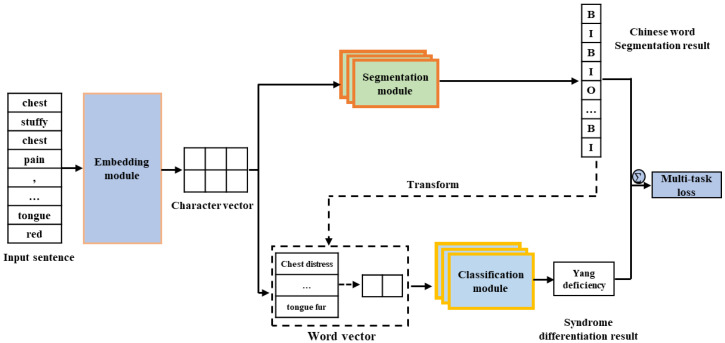
An overview of our framework.

**Figure 2 ijerph-19-05601-f002:**
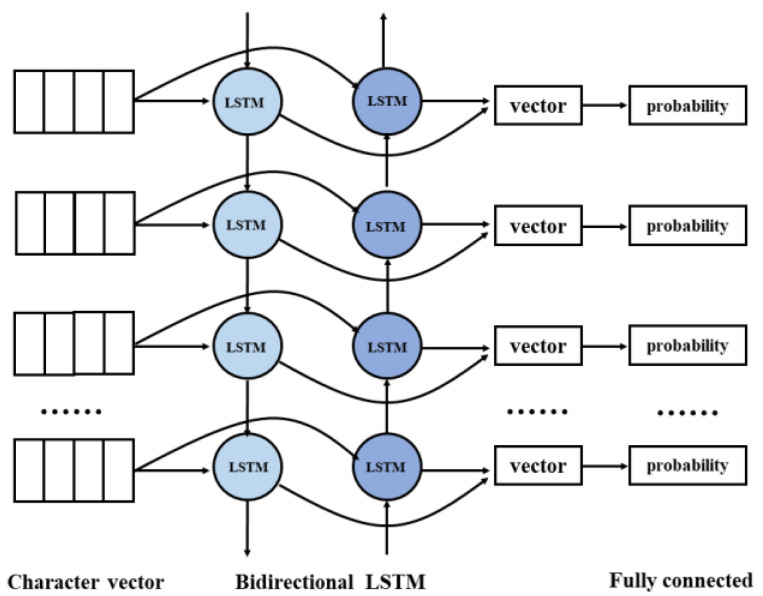
Structure of the segmentation module.

**Figure 3 ijerph-19-05601-f003:**
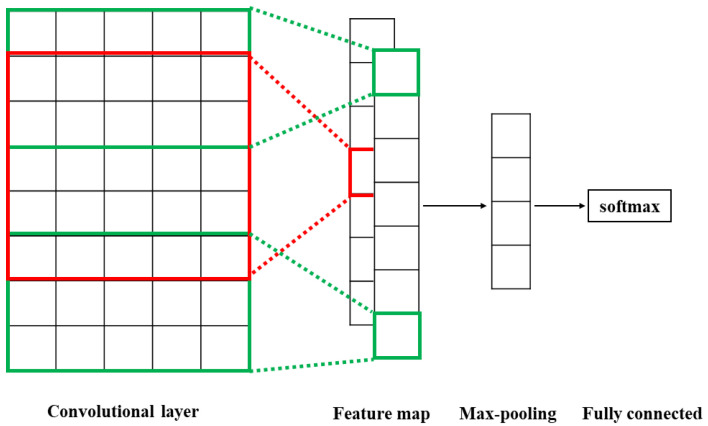
Structure of the classification module.

**Figure 4 ijerph-19-05601-f004:**
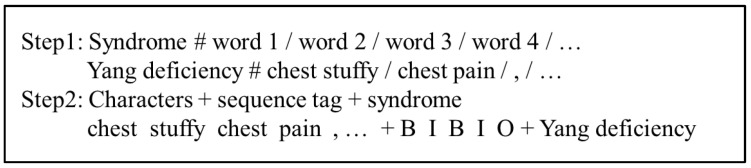
An example of data annotation.

**Figure 5 ijerph-19-05601-f005:**
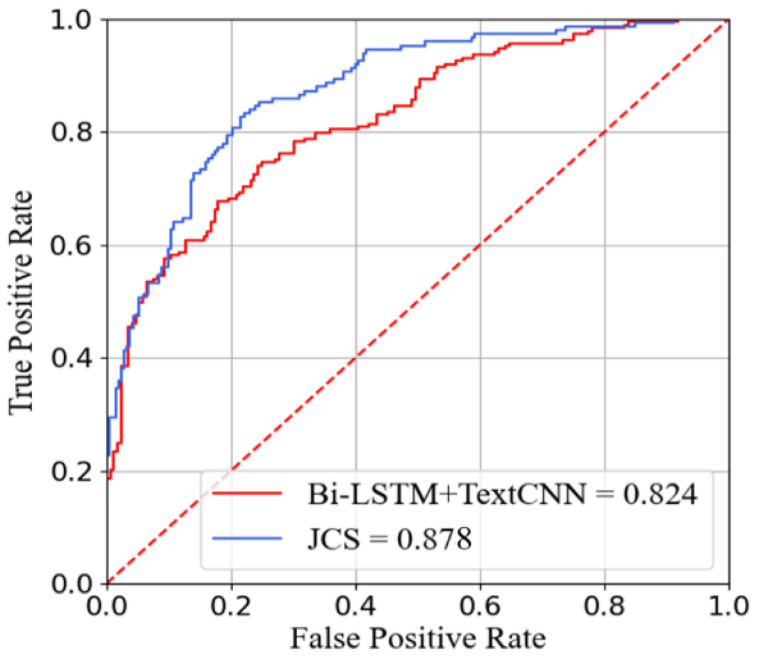
ROC curves of the proposed model and the baseline. The values in the legend represent the AUC values, that is, the area under the ROC curve.

**Table 1 ijerph-19-05601-t001:** Comparative experiments on different loss optimization strategies.

Strategies	Accuracy	Specificity	Sensitivity
Grid Search	0.7995	0.7897	0.8133
Dynamic Weight Averaging	0.7692	0.8364	0.6733
Uncertainty Weighting	0.7830	0.8037	0.7533
Gradient Normalization	0.8022	0.8178	0.7800

**Table 2 ijerph-19-05601-t002:** Comparison between our model and previously proposed models.

Methods	Accuracy	Specificity	Sensitivity
LSTM	0.8752	0.8905	0.7925
Bi-LSTM	0.8536	0.8868	0.7852
Bi-GRU	0.8557	0.8868	0.7778
BERT	0.9464	0.9523	0.9116
RoBERTa	0.7262	0.7066	0.3730
XLNet	0.7080	0.8406	0.2744
JCS	0.9317	0.9436	0.8995

**Table 3 ijerph-19-05601-t003:** Comparison between our model and prevalent methods.

First Stages	Second Stages	Joint/Non-Joint	Accuracy	Specificity	Sensitivity	Time (min)
-	TextCNN	Non-joint	0.7348	0.7073	0.8786	3.5
Bi-LSTM	SVM	Non-joint	0.7521	0.7099	0.7861	14.9
Bi-LSTM	TextRNN	Non-joint	0.6906	0.8671	0.5291	17.5
Bi-LSTM	TextCNN	Non-joint	0.7624	0.8324	0.6984	15.9
JCS		Joint	0.8022	0.8178	0.7800	10

**Table 4 ijerph-19-05601-t004:** Results of comparison with state-of-the-art models.

Methods	Accuracy	Specificity	Sensitivity	AUC
BERT	0.7989	0.7409	0.8647	0.8637
RoBERTa	0.7769	0.7353	0.8135	0.8581
XLNet	0.5455	0.5588	0.5338	0.5394
JCS	0.8022	0.8178	0.7800	0.8780

**Table 5 ijerph-19-05601-t005:** Statistical analysis of the two models.

Models	Accuracy(95% Confidence Interval)		pvalue
Bi-LSTM + TextCNN(non-joint)	0.7624(0.6934, 0.7790)	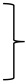	0.0281pvalue<0.05
JCS (joint)	0.8022(0.7486, 0.8314)

**Table 6 ijerph-19-05601-t006:** Comparative experiments on different modules.

Segmentation Module	Classification Module	Joint?	Accuracy	Specificity	Sensitivity
Bi-LSTM	TextCNN	N	0.7624	0.8324	0.6984
		Y	0.8022	0.8178	0.7800
	FastText	N	0.7624	0.7142	0.8150
		Y	0.7665	0.7336	0.8133
Bi-LSTM-CRF	TextCNN	N	0.7541	0.8035	0.7090
		Y	0.7885	0.7733	0.7991
	FastText	N	0.7735	0.8208	0.7302
		Y	0.7885	0.8267	0.7617

## Data Availability

The data presented in this study are available on request from the corresponding author. The data are not publicly available due to it forms part of an ongoing study.

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
