# Peer review of "Multi-Task Joint Learning Model for Chinese Word Segmentation and Syndrome Differentiation in Traditional Chinese Medicine"

_ijerph, 2022, doi:10.3390/ijerph19095601_

Round 1

Reviewer 1 Report

Thanks for the opportunity of reviewing this research. This paper presents a study on multi-task Chinese medicine segmentation and classification for named entity recognition. This work proposes a joint learning framework, under which end-to-end supervision training is carried out. In this work, TextCNN is used to complete the text segmentation task, and Bi-LSTM is used to complete the text classification task. The loss after training the two models is weighted as the final loss to achieve gradient update. Experiments show that the proposed method achieves good results in integrating accuracy, specificity, sensitivity, time, computing power and other factors, which proves the effectiveness of the method.

This work proposes a joint learning framework to solve the segmentation and classification tasks, which effectively solves the needs of the scene. Another positive area of this work is that it used two simple models which could reduce the cost of training and reasoning. The results show that this method has fast reasoning speed and high accuracy. The experimental demonstration of the model performance is sufficient and the conclusion is reliable.

Although this paper has some positive aspects, it still requires significant changes in order to be published. Below are some questions and issues that the authors need to address in the next round:

1. The current version reads like a Computer Science paper, and not a paper targeting the audience of this journal. There are not many connections to the health community. Please strengthen the paper by adding your motivations related to health, and stating how this work can be applied in the health sector. 

2. To follow up on the above point, please also explain the values that this approach can be brought into Traditional Chinese Medicine.

3. On page 3, the caption of Figure 1 is wrongly positioned.

4. On page 4, how are the labels of the data set marked BIO in the sequence obtained? Is it an open source data set? Or did you mark it yourself? It is suggested to introduce the BIO method to facilitate some readers who do not learn it to quickly understand the method and principle.

5. On page 5, the size of the red box is not [3,5]. Is it just a schematic diagram?

6. On page 5, how to determine the weight in Formula 2?

7. On page 5, there are no examples in Section 3.1 data set. It is recommended to add examples to help readers understand the task.

8. On page 6, Section 3.2: why use seven-fold? not ten-fold cross-validation that is commonly used?

9. On page 6 in Section 3.2, the optimizer used is Adam. Please proves that the deep learning library or your code used does not have the problems mentioned on the following web page: https://towardsdatascience.com/why-adamw-matters-736223f31b5d

10. On page 7 in Table 2, when comparing the results with BERT, what are the specific comparison results of time, computing power and the number of parameters?

11. On page 8, what is JSC in Figure 4? Is that your own method/metric?

12. On page 9 in Section 3.6.2, the joint learning model is better than the non-joint model but only the examples are given to illustrate this conclusion (using the word "usually" would be much better).

Reviewer 2 Report

Dear editors and authors, thanks for the opportunity to review this article.

This article describes a multi-task joint learning model. The study described is complex and innovative, and it can be helpful to perform word segmentation on textual medical records. The authors state that they have developed an objective, quantitative, and computer-assisted syndrome differentiation method that can help modernize traditional Chinese medicine. Thus, the research seems valuable and helpful. Nevertheless, some essential aspects of the article perhaps should be revised. These are the following:

One important aspect is that the article's structure could perhaps be revised.

In Proposed Methods, the authors describe that «We seek to improve performance in terms of Chinese word segmentation and syndrome differentiation to analyze medical records in an end-to-end manner». This text seems to describe the main objective of the research. Thus, it seems to fit better at the end of the Introduction.

In Methods, I would invite the authors to begin this section by describing the overall research performed.

In 3. Results/ 3.1. Dataset, the authors write that «The "Essences of Modern Chinese Medical Records of Modern Chinese Medicine" series mainly contains medical records of many famous physicians in TCM in the country [21]. Each medical record in the series was selected by a well-known physician, and its contents are considered reliable in the area. We used records from the fifth and sixth episodes of this series as the original dataset for this study. To ensure validity and accuracy, we preprocessed the text of medical records using text extraction, and solicited three TCM physicians to annotate and inspect the data [22, 23]». I think that this paragraph would fit better in the Methods section.

The same can be applied to subsections 3.2.1, 3.2.2., 3.3.1, 3.4., 3.4.1, 3.5., 3.6., 3.6.1., and other text parts.

I would invite the authors to revise the overall structure of the article. I understand that the method is complex. Due to this, it is essential first to describe the methods clearly and, after, describe the results. Describing methods within the results does not fit the classical article structure and seems confusing for the readers.

Another critical aspect of revising is the Discussion section, which seems too short. It does not seem like a real discussion, as the findings are not compared with other research. For example, the authors state, "Our proposed joint learning model achieves superior performance in TCM Chinese word segmentation and syndrome differentiation tasks. Firstly, in the Chinese word segmentation task, through a comprehensive comparison of performance and space complexity, our proposed model is superior. Meanwhile, in the syndrome differentiation task, our proposed model outperforms other prevalent methods in performance, runtime, and space complexity». Nevertheless, they do not compare their study to other research, and no references are shown. The same can be applied to the second part of the discussion, which is similar.

It is easy to understand that this research is complex and innovative, and perhaps it is not easy to find other similar studies to compare the findings. Nevertheless, the discussion is necessary to obtain the conclusions.

In this sense, the authors write that «The experimental comparisons showed that it was superior to other methods [...]. The experimental comparison showed that our method is superior to prevalent methods [...]. The work here can help modernize TCM through intelligent differentiation». Thus, it would be helpful if they described those other methods mentioned in the discussion and how they obtained their conclusions.

The Limitations section should describe some relevant limitations associated with the study. For example, the software or algorithms may have some intrinsic limitations. The amount of information used for the analyses can add limitations. There is some risk of selection bias that perhaps could be assessed. I want to invite the authors to describe if these are potential limitations or others that may be assessed when considering the external validity of their findings.

In «3.5. tatistical Analysis», it should be «3.5. Statistical Analysis».

Reviewer 3 Report

The study used text classification to model syndrome differentiation for TCM, and use multi-task learning (MTL) and deep learning to accomplish the two challenging tasks of Chinese word segmentation and syndrome differentiation. The topic is interesting but some issues should be addressed to improve the quality of this manuscript.

  1. In the introduction, the first three paragraphs need to add citations to support the statements.
  2. The research gaps were not clearly stated in the introduction. The authors mentioned many works have been done in this research area. What is the novelty and significance of this study? Did MTL outperform other machine learning methods in relevant applications in previous studies? More elaborations about MTL should be provided to readers.
  3. Figure 1 was not clearly explained. What do Yang deficiency and Yin deficiency mean?
  4. In section 3.1., what is a well-known physician? How to define “well-known”? This term is not appropriate. The data inclusion and exclusion criteria are missing. Why was only one physician asked to select the data?  
  5. In Table 2, the method “BERT” is better than the method proposed in this study. Please explain this.
  6. The discussion part should focus more on theoretical contributions and practical implications. I would like to see more in-depth discussions about the findings.
  7. In section 4.2., the limitations of this study were not clearly stated. Also, what are the advanced models used to improve the performance of the model? Why did the authors not use the advanced models in this study instead?
  8. There are language issues in the manuscript. Proofreading should be conducted.

Round 2

Reviewer 3 Report

The authors did a good job in addressing my comments. I have no further comments.